# Ultrasound-Guided Interventions in the Biliary System

**DOI:** 10.3390/diagnostics14040403

**Published:** 2024-02-12

**Authors:** Thomas Müller, Barbara Braden

**Affiliations:** 1Medizinische Klinik II, St. Josefs-Hospital, Beethovenstraße 20, 65189 Wiesbaden, Germany; 2Medizinische Klinik B, Universitätsklinikum Münster, Albert-Schweitzer-Campus 1, 48149 Münster, Germany

**Keywords:** percutaneous biliary drainage, percutaneous gall bladder drainage, ultrasound guidance

## Abstract

Ultrasound guidance in biliary interventions has become the standard tool to facilitate percutaneous biliary drainage as well as percutaneous gall bladder drainage. Monitoring of the needle tip whilst penetrating the tissue in real time using ultrasound allows precise manoeuvres and exact targeting without radiation exposure. Without the need for fluoroscopy, ultrasound-guided drainage procedures can be performed bedside as a sometimes life-saving procedure in patients with severe cholangitis/cholecystitis when they are critically ill in intensive care units and cannot be transported to a fluoroscopy suite. This article describes the current data background and guidelines and focuses on specific sonographic aspects of both the procedures of percutaneous biliary drainage and gallbladder drainage.

## 1. Introduction

Interventions in the biliary system might target the bile ducts or the gall bladder. The two most commonly practised procedures are percutaneous transhepatic biliary drainage (PTBD) and percutaneous (transhepatic) gallbladder drainage (PGBD) [1]. Ultrasound (US) guidance of punctures and drainage placements offers some apparent advantages over “blind” punctures and fluoroscopic or computed tomography guidance. US, whether applied percutaneously or endoscopically, is the only modality that allows continuous and real-time monitoring of the needle trajectory and the advancement of the needle tip through the penetrated tissue. The reduction in or even avoidance of ionising radiation is another major benefit for the patient and doctor. Therefore, not using ultrasound might need justification.

This article will focus on the special sonographic aspects of biliary interventions. We searched the Cochrane Library, PubMed (including Medline), and Google Scholar for relevant studies published between 2000 and 2023. Reported practical aspects are based on the authors’ comprehensive experience in US-guided biliary interventions.

## 2. Biliary Drainage

Endoscopic biliary drainage (EBD) using metal and plastic stents inserted by endoscopic retrograde cholangiography (ERC) is the standard procedure for draining the common bile duct and the intrahepatic biliary system. PTBD is an alternative approach if the major duodenal papilla is not accessible or the endoscopic approach via the papilla has failed. This might be due to postsurgical changes in the anatomy (Roux-en-Y operation and others) or if a stenosis of the biliary system cannot be overcome by the endoscopic approach [2,3,4]. PTBD does not seem to be inferior to EBD [5,6,7]. Endoscopic ultrasound-guided biliary drainage (EUS-BD) is an increasingly used third method of accessing the biliary system when ERC fails to achieve biliary drainage, but so far, this is only available in specialised centres [8,9,10]. Thus, the choice of the appropriate drainage method when EBD fails or is not applicable is “based on the location of the obstruction, the purpose of drainage (as a preoperative procedure or a palliative treatment) and level of experience in biliary drainage at individual treatment centers” [5].

Indications for PTBD include palliative treatment (e.g., bile drainage in unresectable metastases, and biliary, papillary, or pancreatic carcinoma) as well as access to minimally invasive procedures in benign diseases (dilation of benign strictures, percutaneous stone removal, drainage in cholangitis, preparing a channel for direct cholangioscopy, etc.) [2,4,11,12,13,14].

From its beginning in the 1930s and for many decades, PTBD used to be a solely fluoroscopic procedure [15]. With the increasing use of ultrasound as a diagnostic tool in the late 1970s and 1980s, US guidance for the initial puncture of bile ducts was recommended by a few radiologists. It seemed desirable to have a guidance method at hand that provided continuous monitoring of the needle’s tip and tracking through the tissues in real time. From a theoretical point of view, continuous guidance was supposed to reduce the number of needle punctures, fluoroscopy time, total procedure time, and complication rates [16,17,18,19]. Targeting bile ducts with a needle is a challenging procedure and often requires multiple passes before it is successfully achieved. Each pass increases the risk of bleeding. Ultrasound provides excellent spatial and temporal resolution and, therefore, will facilitate access to the biliary system. Over the years, US guidance has become more and more common.

After an initial puncture under ultrasound guidance, the procedure is usually continued under fluoroscopic imaging [3,20,21,22,23]. As long as only external drainage is to be obtained, US guidance alone without fluoroscopy is feasible and sufficient [17,24,25]. Without fluoroscopy, ultrasound-guided PTBD, performed at the bedside in intensive care, can be life-saving to critically ill patients who cannot be transported to a fluoroscopy suite [24].

One case series illustrates the feasibility of also achieving external–internal drainage using contrast-enhanced ultrasound. A diluted ultrasound contrast agent was administered via the percutaneously placed needle for visualising the biliary system. However, fluoroscopy was necessary for depicting the guide wire on its way through the stenosis [26]. Intrabiliary application of a diluted contrast agent may be useful for depicting drainage tips and identifying stenoses [27,28].

Fusion imaging might be used as well for the initial puncture [29], but the clinical relevance of this method, combining US and CT or MRI imaging, cannot currently be assessed.

### 2.1. Data Background

While since 1980, numerous feasibility studies on US guidance have been published [21,22,23,24,25,30,31], only very few studies compared US-guided vs. fluoroscopically guided initial puncture in PTBD retrospectively [32,33]. Prospective comparative studies have not been published.

An older Russian study (including a total of 89 patients) favours US guidance due to a decrease in radiation time, higher technical success, a lower number of needle passes, and decreased complication rates in the US group [32]. A larger German retrospective analysis (*n* = 251), on the other hand, showed no differences in the total complication rate, fluoroscopy time, and success rates [33]. A recent systematic review and meta-analysis of bleeding rates in PTBD found no reduction in the US group [34].

Other authors compare their results from US-guided PTBD to data from the literature and suggest lower complication rates [30] or a lower radiation time [23,31].

### 2.2. How to Perform US-Guided PTBD

US-guided PTBD is performed under sterile conditions, including protective clothing, skin disinfection, sterile covers on keyboards and probes, and sterile drapes [35,36]. Local anaesthesia of the parietal peritoneum and liver capsule is mandatory. We usually do not administer sedative medication in order to maintain a patient’s cooperation (particularly for breath holding).

The biliary system can be punctured in the right or left liver lobe, depending on the level of obstruction or medical needs. In dilated intrahepatic ducts, accessing the bile ducts with a needle and advancing a wire with the subsequent placement of drainage is easier, and adverse events are fewer than in nondilated ducts. It is also possible to access nondilated ducts, but ideally, a duct with a diameter of ≥3 mm should be aimed for. As bile ducts run in parallel to portal veins, nondilated bile ducts can be found by using colour Doppler and heading for a portal vessel visualised thereby—the so-called parallel technique [20,31,37,38,39]. Ideally, the angle between the duct and needle should be less than 90 degrees [40].

PTBD is commonly performed in the so-called “in plane technique” (Figure 1a,b).

The needle is inserted at the narrow side of the transducer and advanced in the US plane. It can thereby be depicted on its whole trajectory. For a successful puncture, the interventionalist has to keep the needle shaft, tip, and target (bile duct) on the image at all times. Losing the tip or the target being out of sight will obviously prohibit a successful puncture. Not having the needle shaft in sight but seeing the tip and target means the further trajectory of the needle will miss the bile duct. Careful tiny corrections, either of the probe orientation or needle direction, must then be undertaken to have all three in sight during the procedure.

A needle-holding attachment (Figure 2) can be used. It will facilitate keeping all elements “in plane”.

However, the “free hand” technique allows for every puncture angle and is, therefore, preferred by us. The use of colour Doppler will enable the doctor to avoid blood vessels and identify smaller bile ducts (Figure 3a,b).

In cases of extrahepatic obstruction, mostly the right liver lobe will be addressed. It is common practice to head for the peripheral branches in Segment V (cf. Figure 3a,b), as from there to the common bile duct, the biliary branches form a harmonic arch, and the forward movement of the guide wire seems unpretentious [3]. However, US guidance allows the puncture of any biliary duct of the right or left liver lobe that can be depicted well (Figure 4a–c).

If peripheral bile ducts are not dilated and puncture is difficult, sometimes, targeting the central bile ducts and even the common bile duct close to the hilum are the only options to place drainage. When trying to puncture the common bile duct, it is sensible to enter it as proximal as possible to avoid biliary leakage after drain removal. The needle tract should point towards the distal common bile duct (CBD) to facilitate guide-wire advancement (Figure 5).

Usually, a Chiba needle of 21 G is used for the primary puncture. These needles can be depicted quite well in US given that the angle is pointed. Once the targeted bile duct has been accessed, the procedure is continued under fluoroscopic guidance by injecting a small amount of contrast agent to confirm correct needle placement and to visualise the biliary system fluoroscopically (Figure 6). 

The application of diluted US contrast agents such as SonoVue^®^ can confirm correct needle placement as well (Figure 7) but will worsen guide-wire visibility thereafter.

Then, a thin 0.018” guide wire is placed through the needle into the bile duct. Subsequently, a plastic 5F catheter is inserted over the wire. Thin metal guide wires are clearly visible on the US (Figure 8), but usually, the procedure is continued under fluoroscopic guidance to advance the wire across the stricture into the duodenum. 

Subsequently, exchanging to larger and more stable 0.035” guide wires allows stepwise bouginage of the tract and the placement of drainage [3,14,41,42]. Ideally, drainage should be placed with side holes above and below the stenosis and the tip in the duodenum for combined extern–intern drainage. If the placement across the stricture into the duodenum fails, external drainage can be placed. 

If external drainage is initially planned, it can be placed under US guidance alone; fluoroscopy and cholangiography are not necessary. External drainage has a higher risk of dislocation; therefore, a pigtail catheter should be chosen to reduce this risk. A thread lock that prevents the uncoiling of the pigtail may be helpful, too. 

If the bile has been drained for 1–2 days via the external catheter, and the intrahepatic ducts have been decompressed, it is often easier to advance the wire successfully into the duodenum to internalise the drainage.

Special consideration should be given to the distance between the skin and the first side hole of drainage to avoid leakage into the peritoneum and to ensure that the first side hole is above the stricture.

## 3. Percutaneous Gall Bladder Drainage (PGBD or Percutaneous Cholecystostomy)

As laparoscopic cholecystectomy is the standard procedure for the treatment of acute cholecystitis, most patients will undergo surgery. However, a considerable number of patients are not eligible for surgery. PGBD, endoscopic ultrasound-guided gallbladder drainage (EUS-GBD), and endoscopic transpapillary gallbladder drainage (ETGBD) are minimally invasive alternatives that can be considered [43,44,45]. While endoscopic approaches require equipment and expertise, PGBD is relatively easy to perform and is recommended for these patients in various guidelines [45,46,47,48,49,50]. Ultrasound is widely available, including in less specialised centres. Procedure descriptions for ultrasound-guided PGBD date back to the 1970s [51].

### 3.1. Data Background

There are countless feasibility studies and retrospective data analyses on PGBD and just one recent randomised controlled trial (RCT) comparing PGBD to cholecystectomy [52]. Indications for PGBD vary considerably. In some studies, PGBD was reserved for patients not eligible for surgery; in others, it was offered as an alternative treatment option; and in other centres, it was a bridging therapy to surgery. The results can hardly be compared [52,53,54,55,56,57,58,59,60,61]. In general, patients receiving PGBD have more comorbidities or critical illnesses than the ones allocated for surgery [54,62] or for conservative treatment [63], most probably contributing to the higher complication rates.

Technical success is reported to be 100% [59,64]. Retrospective studies report clinical success rates of around 50% [55,56,58] for patients treated definitively with PGBD. However, often, the observation times were short or not reported.

One study reports a readmission rate of 25.3%, with a median follow-up of 1.6 years [65]. In some publications, 40–50% of patients treated by PGBD subsequently underwent emergency or elective cholecystectomy [52,55].

In 2013, a systematic Cochrane review including two trials found no differences in the mortality and morbidity between patients who underwent surgery vs. PGBD but was “unable to determine the role of percutaneous cholecystostomy in the clinical management of high-risk surgical patients with acute cholecystitis” [66]. Other and more recent meta-analyses, with one including nonrandomised trials, claimed PGBD to be inferior to cholecystectomy in terms of mortality, reinterventions, and hospital readmissions, even in high-risk patients [53,67].

The Society of Interventional Radiology published quality improvement standards for PGBD, recommending US guidance for GB puncture [1].

Not included in the recommendations in the guidelines, antibiotics alone, simple (repeated) aspiration, and observation only are other treatment options [62,63,65,68,69,70,71,72].

The most frequent complication of PGBD is drainage dislocation, with reported rates from 7.4% to almost 30% [54,55,73].

An RCT comparing PGBD and EUS-GBD in patients not undergoing subsequent cholecystectomy showed reduced adverse event rates, less reinterventions after 30 days, and a lower number of unplanned readmissions in the EUS-GBD group [64]. A recently published prospective trial compared patients with acute cholecystitis who underwent EUS-GBD or PGBD, followed by an attempted cholecystectomy. There were no significant differences in adverse event rates related to the drainage procedure or subsequent surgical and technical success but a decreased operative time, shorter time to symptom resolution, and length of postsurgical stay in the EUS-GBD group [59].

A systematic review and meta-analysis analysed eleven studies including 1155 patients. The outcomes were technical and clinical success, adverse events, recurrent cholecystitis, reintervention, and hospital readmission. There was no difference between PGBD and EUS-GBD in all the evaluated outcomes. Only the LAMS subgroup was associated with lower rates of adverse events, recurrent cholecystitis, and hospital readmission [74]. 

Another systematic review and meta-analysis published in the same year and analysing almost the same data (1136 patients from eleven studies) focused on technical success, clinical success, and adverse events as outcomes. EUS-GBD in 9 of 11 studies used LAMS only for drainage and had significantly better technical success, fewer adverse events, and lower reintervention rates than PGBD. No difference in clinical success or readmission rate was found [75].

### 3.2. How to Perform US-Guided PGBD

US-guided PGBD is performed under sterile conditions, including protective clothing, skin disinfection, sterile covers on keyboards and probes, and sterile drapes [35,36]. Local anaesthesia of the parietal peritoneum and liver capsule is mandatory. We usually do not administer sedative medication in order to maintain a patient’s cooperation (particularly for breath holding).

Most authors recommend a transhepatic approach in order to prevent biliary leakage to the abdominal cavity during drainage or after removal of drainage [43,55,56,76,77] (Figure 9). 

The consideration is that a perforation in the hepatic surface of the gallbladder will be covered by the liver tissue after the removal of the drain. However, retrospective analyses comparing the transhepatic route with the transperitoneal (i.e., nontranshepatic) access (Figure 10) report no differences in the complication rates [58,78,79].

Again, the “in plane” approach is recommended to depict the needle in real time and continuously on its way. This is mandatory because iatrogenic perforation of the posterior gallbladder wall must be avoided. Both the drainage techniques, Seldinger (Figure 11a–c) and trocar (direct puncture, cf. Figure 10), may be used, depending on the physician’s preferences [43,58]. 

We recommend the use of 8F or 10F pigtail drainage. If it is inserted using Seldinger’s technique, dilatation is not necessary. If the guide wire is confirmed to be in the correct place and is secured well, drainage placement can be performed without US visualisation [43,76].

As tube displacement is a relevant problem, we recommend the use of thread-locked catheters, which are supposed to keep the pigtail in its curled shape, thereby preventing dislocation.

### 3.3. Duration of Drainage

The aims of PGBD are the relief of symptoms and the disappearance of inflammation. If these criteria are met, the drainage can be removed. After pre-existing GB perforation, the integrity of the gallbladder wall should be documented. This can be performed very well by the injection of US-contrast agents (Figure 12 and Figure 13) [80].

After the injection of one drop of SonoVue^®^, e.g., diluted in 10 mL sodium chloride 0.9% via the drain, gallbladder perforation and leakage can be excluded when only the drainage and the gall bladder lumen show enhancement, and no extravasation is observed.

Too early removal of the gallbladder drainage might cause the recurrence of cholecystitis. Leaving the drainage in place for too long might lead to persisting biliary leakage, as known from PTBD. Determining the perfect time for removal can be challenging.

In the literature, the mean time of drainage therapy surprisingly ranges from 6 days [65], 12 days [58], 16 days [76], between 1 and 2 months [78,81,82] to almost 3 months [56,83]. In one publication, up to ten tube changes or drain replacements were reported [56]. These largely varying drainage durations reflect different therapeutic approaches. Some authors see drainage therapy as a tool in primarily curing acute cholecystitis, while others believe continued drainage therapy is necessary to prevent recurring inflammation, especially in calculous cholecystitis and patients definitely unfit for surgery.

In our experience, about six days of drainage are appropriate in most cases to cure acute cholecystitis.

Some authors recommend a “maturation” of the tract in order to prevent biliary leakage after removal [61,76,84]. In our experience, early drainage removal and quick channel closure should be therapeutic aims. A maturated tract might lead to a persistent biliocutaneous fistula.

A systematic review showed no correlation between overall mortality, biliary mortality, overall morbidity, or the disease recurrence rate and the timing of catheter removal [85].

The decision on the extraction of gallbladder drainage should be based on improvement in the inflammatory and clinical parameters [76] rather than on the sonographic appearance of the gallbladder because the ultrasound assessment of the emptied gallbladder is difficult. Some authors claim that cystic duct patency should be proven as a precondition for drainage removal [55,86,87]; other authors report no benefit of routine cholangiography before drain removal [88].

Whether PGBD can be seen as a definitive treatment or whether subsequent endoscopic or surgical interventions could be considered has to be decided on an individual basis [43,61].

## 4. Percutaneous Gallbladder Aspiration (PGBA)

A number of feasibility studies and retrospective analyses report on single or repeated percutaneous transhepatic gallbladder aspiration as a treatment option in acute cholecystitis. A noncontrolled prospective trial with 33 patients reported a success rate of 76% for a single aspiration. The mean hospital stay of patients with successful aspiration was three days [70]. An older randomised controlled trial compared PGBD and PGBA in 30 patients and 23 patients, respectively. There was a significant difference in good clinical responses, which could be obtained in 27 patients (90%) of the PGBD group and in 14 patients (61%) of the PGBA group [89].

As the relief of symptoms occurs rapidly after aspiration and not all patients with acute cholecystitis have positive bile cultures, gallbladder decompression seems to be a therapeutic principle on its own [72]. In our institutions, we use PGBA as a bridging therapy in patients who are expected to be fit for surgery in a short while or in palliative situations.

### How to Perform US-Guided PGBA

The procedure is similar to the first step of PGBD. We also recommend a transhepatic approach, but the transperitoneal route is feasible as well (cf. Figure 9a and Figure 10). We usually use a simple 22G needle with an extension tube, a three-way stopcock, and a 10 mL syringe. Under continuous US control, the gallbladder is evacuated, and bile is sent for microbiological testing.

## 5. Percutaneous Cholecystoenteric Anastomosis

In a very recent preliminary study, a percutaneous approach was used to create a new anastomosis between the gallbladder and the adjacent duodenum using a lumen-apposing metal stent (LAMS) to provide long-term drainage for patients unfit for surgery [90]. Technical success was achieved in 100% of the 14 patients. In 12 patients (86%), the existing cholecystostomy tube could be removed after the insertion of the LAMS.

For internalisation of the gallbladder drainage, the existing cholecystostomy tube was replaced by a 10-F sheath. Through this 10F sheath, an 18-gauge trocar needle was advanced into the gallbladder. Under fluoroscopic guidance, the trocar needle was then used to puncture through the posterior wall of the gallbladder and into the lateral sidewall of the second part of the duodenum distended with CO_2_ via a nasoenteral tube. After exchange to a stiff wire, a 12F sheath was advanced into the duodenum, allowing the fully covered LAMS (10.8F) to follow through. The distal flange of the stent was then deployed within the duodenum and apposed to the wall. The proximal flange has to be deployed inside the gallbladder. The procedure requires precise opening of the flanges for correct positioning. A final contrast-medium injection can then confirm the correct stent placement with drainage from the gallbladder to the duodenum through the stent.

## 6. Discussion

### 6.1. Percutaneous Biliary Drainage

Many radiologists traditionally perform fluoroscopically guided PTBD. US guidance is relatively “new”. In the largest comparative study, 207 fluoroscopically guided PTBDs and 44 US-guided PTBDs were performed [33]. The comparison of new methods to well-established and often practised approaches carries the risk of “experience bias”. As the performance and success rate of US-guided punctures generally improve with training [91,92,93,94], it would have been of interest to know about the experience of the doctors performing US-guided PTBD. However, the published studies could not provide that specific information. 

In conclusion, the data quality of studies on US-guided PTBD is low, and the results are heterogeneous. Expected advantages, like a reduction in the number of needle punctures, fluoroscopy time, total procedure time, and complication rates, as postulated already in the 1980s, have not yet been proven.

However, despite the absolute lack of reliable statistical data and most probably due to its obvious advantages, US guidance of the initial puncture in PTBD is the recommended standard approach in a considerable number of relevant international guidelines [1,46,95,96,97], while others still do not recommend or even mention US as a guidance tool [10,13].

### 6.2. Percutaneous Gallbladder Drainage

Taking into consideration the diversity of allocation criteria for PGBD, success and recurrence rates have to be interpreted cautiously. PGBD is considered a rescue therapy for patients not eligible for surgery; therefore, rates of adverse events should not be compared to those of the standard therapy (cholecystectomy) but to adverse event rates after EUS-GBD and ETGBD [44]. The data analysed in recent meta-analyses are predominantly from retrospective studies. Overall, the complication rates of these three drainage options, PGBD, EUS-GBD and ETGBD, seem to be similar. However, PGBD results in a higher need for reinterventions and carries a higher risk of unplanned readmissions [59,64,73,74,75,98,99]. Fitness for surgical options should, therefore, be re-evaluated after clinical improvement. 

The choice of drainage method will depend on local expertise and the availability of the different drainage techniques [73]. A sophisticated treatment algorithm for patients with acute cholecystitis and varied surgical risk was recently proposed by a group from Orlando [100].

Due to different reimbursement practices and globally varying prices, it is difficult to compare the economic costs of PGBD and EUS-GBD across different healthcare systems. The sheer cost of LAMS used for EUS-GBD is roughly 30–100 times higher than those of drains used for PGBD.

### 6.3. Percutaneous Gallbladder Aspiration

Currently, there is no evidence for PGBA in the treatment of acute cholecystitis. However, it is feasible and may serve as a rescue treatment in patients not eligible for surgery or alternative procedures, such as PGBD, EUS-GBD, or ETGBD.

### 6.4. Percutaneous Cholecystoenteric Anastomosis

Taking into consideration the very early clinical experience, percutaneous LAMS placement with the creation of a cholecystoenteric anastomosis appeared a technically feasible, safe, and effective procedure to achieve internal gallbladder drainage in patients with indwelling cholecystostomy tubes who are not candidates for surgical cholecystectomy. Long-term data are lacking, and further randomised studies comparing EUS-guided and percutaneously placed LAMS drainage of the gallbladder would be highly desirable.

## 7. Conclusions

1. Ultrasound guidance of the initial puncture in PTBD is standard in many international guidelines despite a lack of evidence;

2. An in-plane technique should be used for targeting bile ducts or gallbladders;

3. When external drainage is sufficient, PTBD can be performed under US guidance alone;

4. Laparoscopic cholecystectomy is the standard procedure in acute cholecystitis and is superior to PGBD;

5. US-guided PGBD is a valuable option for patients not eligible for surgery and might even be the definitive therapeutic option;

6. A transhepatic or transperitoneal approach is reasonable;

7. Short-term PGBD for about a week is often sufficient to cure acute cholecystitis;

8. US-guided PTBD and PGBD can be performed bedside without fluoroscopy in critically ill patients;

9. Contrast-enhanced US can be used for the confirmation of correct needle placement and leak tightness of the gallbladder.

## 8. Future Directions

As US guidance of PTBD is recommended in many international guidelines despite a significant lack of randomised controlled trials vs. fluoroscopic guidance, it is doubtful whether future research comparing these guidance methods will be initiated. The assumed advantages of US guidance have obviously already convinced leading experts in this field.

As several meta-analyses claim the benefits of laparoscopic surgery over PGBD, even in critically ill patients, a standardised definition of criteria for what are considered contraindications for the operative procedure seems desirable.

Expertise in EUS-guided interventions is increasing and will become more widespread. The numbers of EUS-GBD using LAMS and transgastric EUS-BD will most probably increase. Therefore, US-guided PGBD might play a decreasing role in future in developed countries with well-equipped healthcare systems. 

## Figures and Tables

**Figure 1 diagnostics-14-00403-f001:**
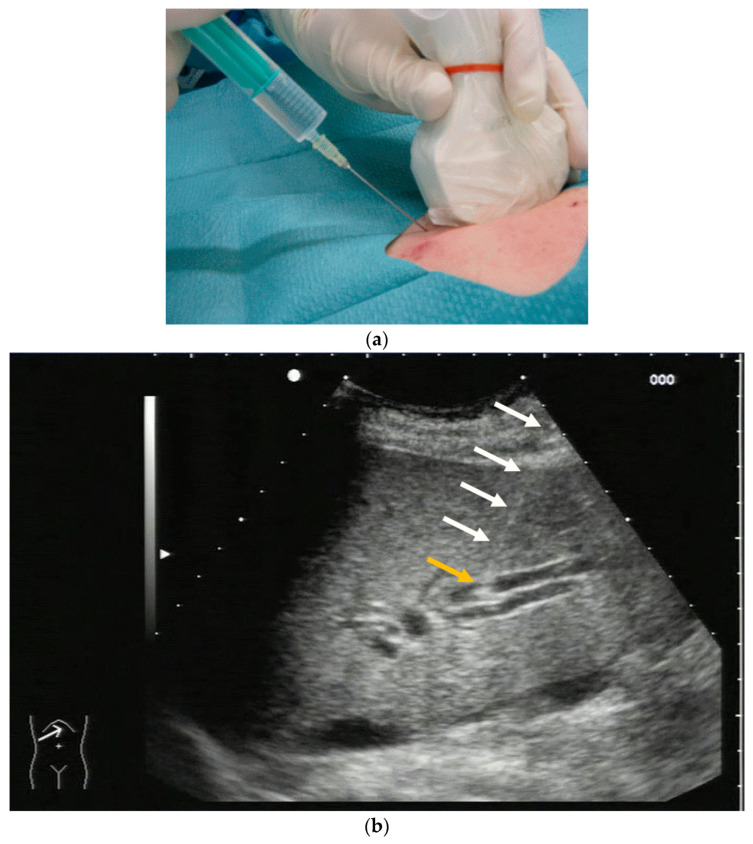
(**a**) Puncture in plane. The needle is inserted at the centre of the small side of the transducer; (**b**) Needle tip (yellow arrow), body (white arrows), and target (bile duct) are depicted in one plane. The needle tip is kept in plane by very small movements of the transducer.

**Figure 2 diagnostics-14-00403-f002:**
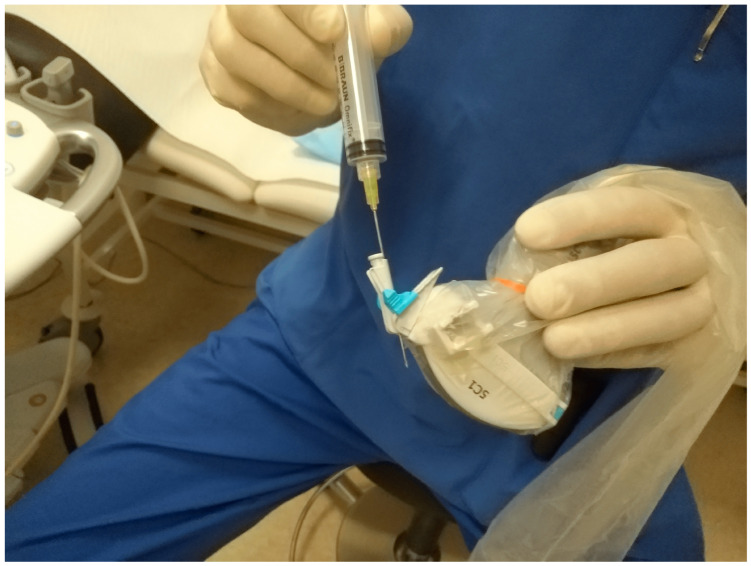
Needle attachment. Up to four given puncture angles can be chosen depending on the US machine manufacturer.

**Figure 3 diagnostics-14-00403-f003:**
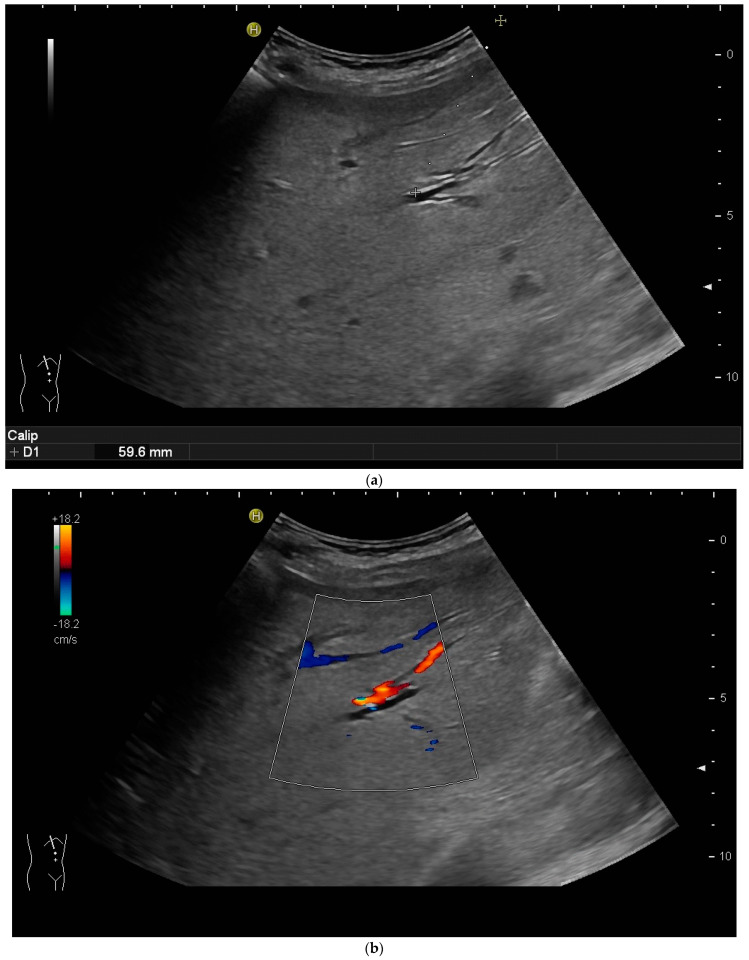
(**a**) Hypoechoic tubular structures in Segment V are visualised in an intercostal view; (**b**) Colour Doppler helps to differentiate between bile ducts and blood vessels.

**Figure 4 diagnostics-14-00403-f004:**
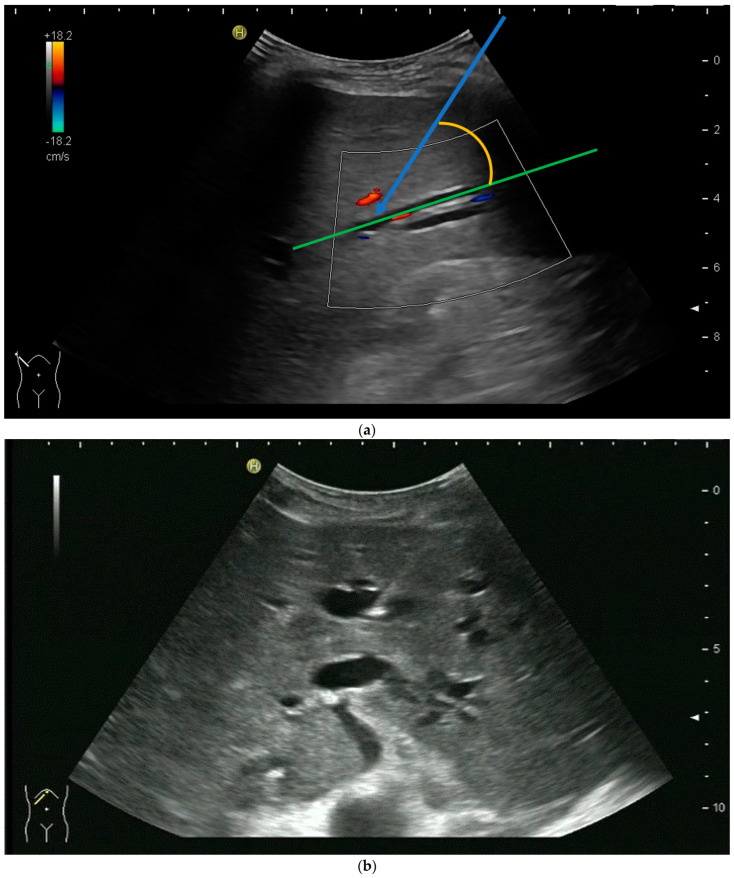
(**a**) A pointed angle between the needle trajectory and a bile duct of Segment V/VI is given; (**b**) Puncture of a dilated bile duct in Segment IV at a suitable angle. A guide wire with a bended tip should be used to facilitate the correct advancement; (**c**) A guide wire is placed in a dilated intrahepatic bile duct in Segment III.

**Figure 5 diagnostics-14-00403-f005:**
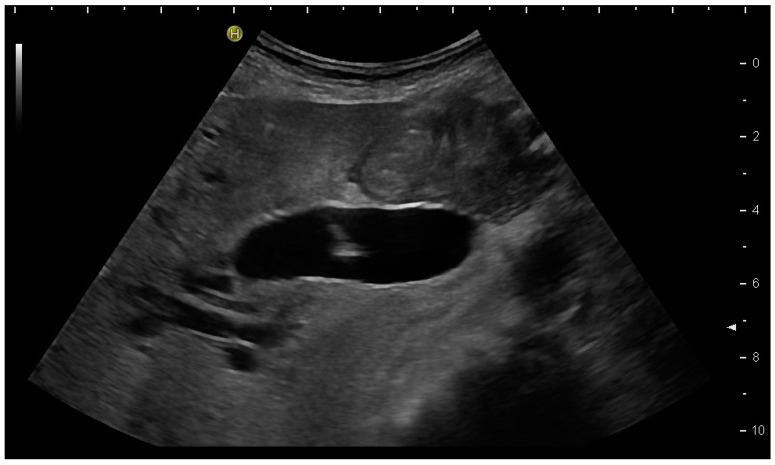
Puncture of the CBD in a patient with pancreatic carcinoma. The tip of the needle is seen in the dilated CBD. Pigtail drainage was placed in the CBD, and ten days later, internal drainage by a self-expanding metal stent was placed using a rendezvous technique (percutaneous transpapillary guide-wire advancement and endoscopic stent placement over the wire).

**Figure 6 diagnostics-14-00403-f006:**
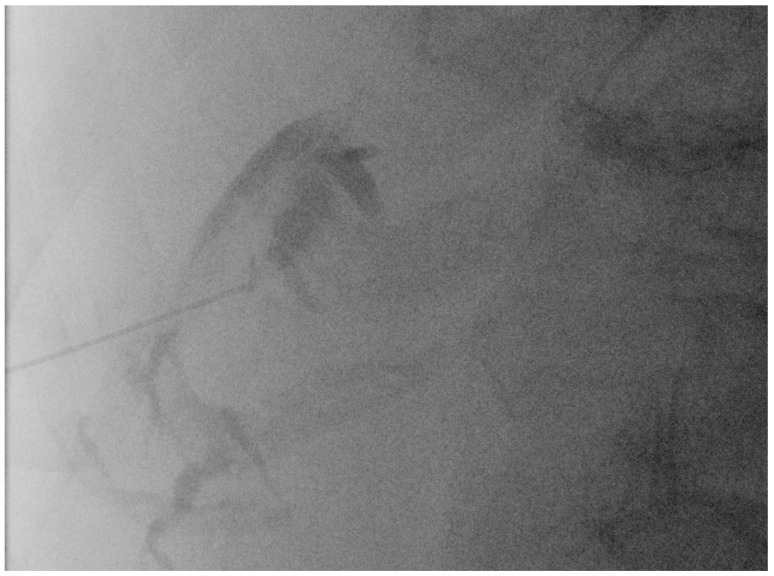
Fluoroscopic confirmation of correct US-guided placement of the 24G needle.

**Figure 7 diagnostics-14-00403-f007:**
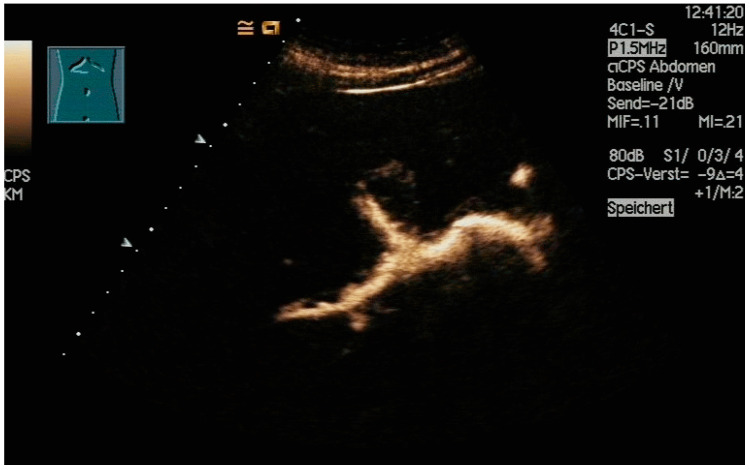
After injection of one drop of SonoVue^®^ diluted in 10 mL sodium chloride 0.9%, the dilated intrahepatic bile ducts are depicted, and correct needle placement is confirmed. However, subsequent monitoring of the guide wires will be hindered.

**Figure 8 diagnostics-14-00403-f008:**
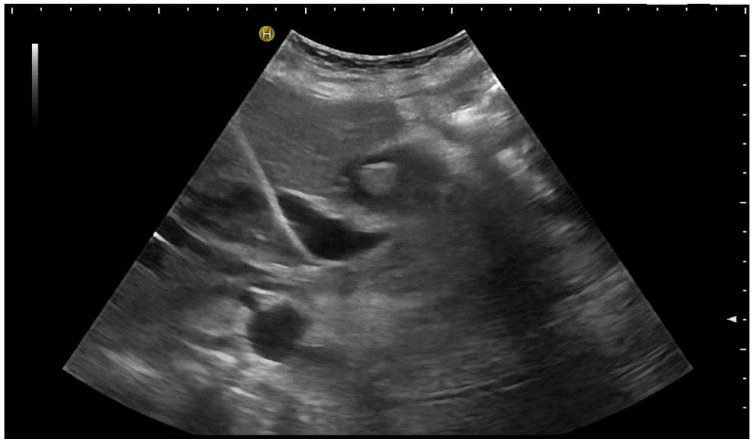
The 0.018” guide wire is very well-depicted by US. The 5F plastic catheter will subsequently be placed; however, there is poor sonographic visibility.

**Figure 9 diagnostics-14-00403-f009:**
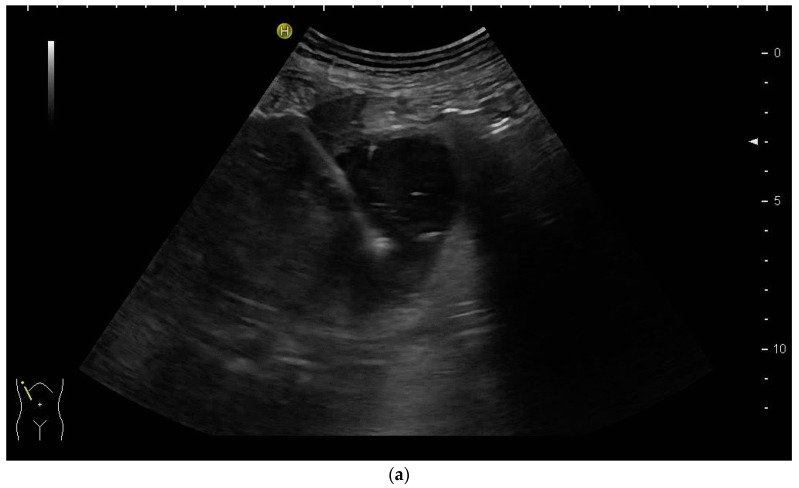
(**a**) Transhepatic puncture of an inflamed gall bladder in the in-plane technique. The Chiba needle is depicted well; (**b**) A few days later, laparoscopic cholecystectomy was performed. The drainage ran through Segment V of the liver. It was removed without provoking biliary leakage from the liver parenchyma.

**Figure 10 diagnostics-14-00403-f010:**
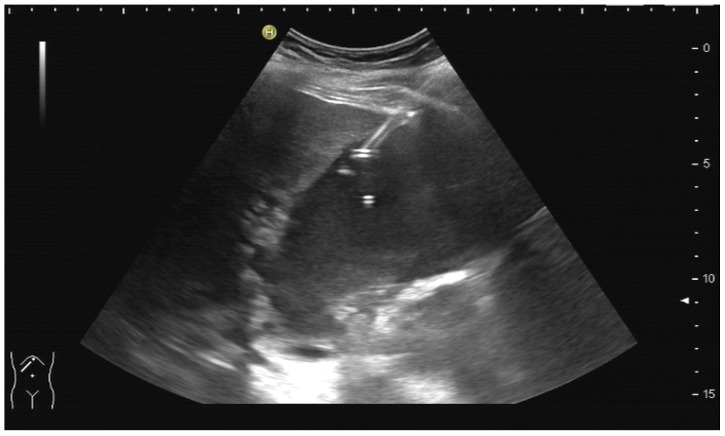
PGBD without passing the liver and using the trocar technique: straightened drainage passes right below the lower liver margin into the gallbladder.

**Figure 11 diagnostics-14-00403-f011:**
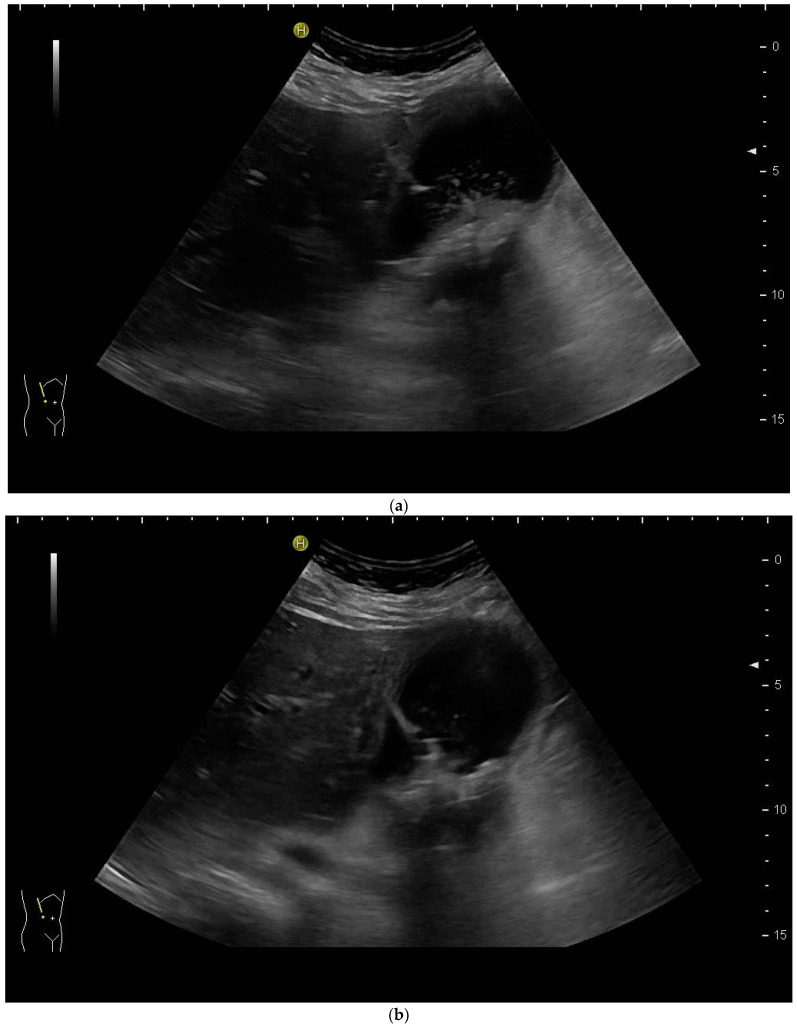
When using the Seldinger technique, first a Chiba needle is placed in the gallbladder (**a**), through which a 0.035” guide wire with bended tip is inserted (**b**). Plastic pigtail drainage (**c**) is then advanced over the guide wire.

**Figure 12 diagnostics-14-00403-f012:**
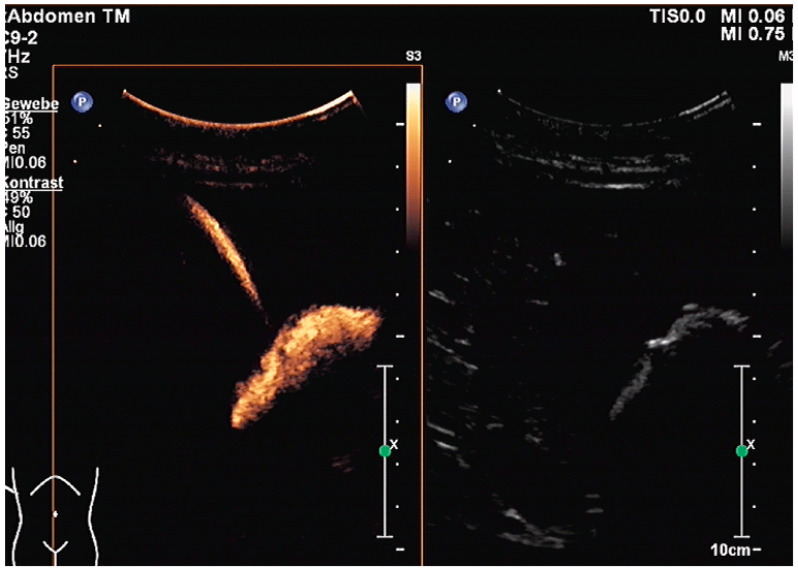
After injection of one drop of SonoVue^®^ diluted in 10 mL sodium chloride 0.9% via drainage, gallbladder perforation and leakage can be excluded. Only the drain and the gall bladder lumen show enhancement (left side: low-MI-mode, right side: conventional grey-scale image).

**Figure 13 diagnostics-14-00403-f013:**
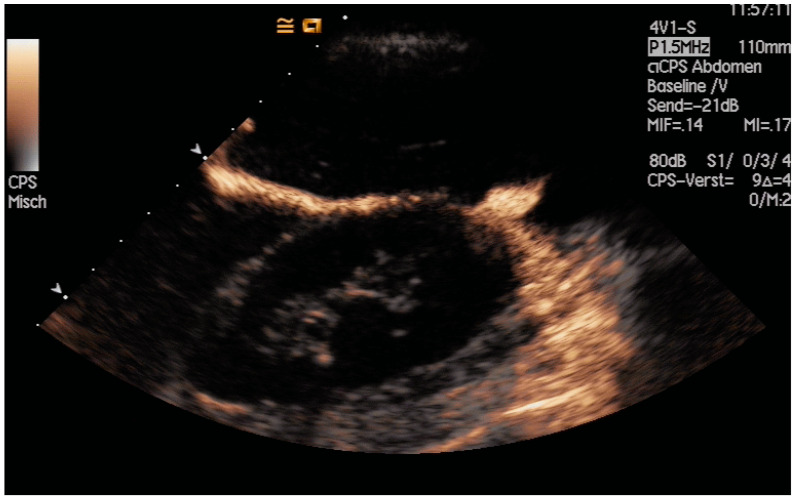
Gall bladder perforation. After injection of diluted US-contrast agent via the drain, contrast media appears at the lower liver surface. The drain was left in place for a few more days until surgery could be performed.

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
