# Peer review of "Ultrasound-Guided Interventions in the Biliary System"

_diagnostics, 2024, doi:10.3390/diagnostics14040403_

Round 1

Reviewer 1 Report

Comments and Suggestions for Authors

Dear authors

Congratulations on this review article, which is interesting and instructive for colleagues in clinical practice. However, I propose the following changes.

In line with the journal's recommendations, I suggest changing the structure of the article. Part of the discussion and future directions are missing. 

  • Review: Reviews offer a comprehensive analysis of the existing literature within a field of study, identifying current gaps or problems. They should be critical and constructive and provide recommendations for future research. No new, unpublished data should be presented. The structure can include an Abstract, Keywords, Introduction, Relevant Sections, Discussion, Conclusions, and Future Directions, with a suggested minimum word count of 4000 words. Best regards

  •  

Author Response

Dear Madam, dear Sir,

We appreciate your suggestion and changed the article’s structure according to your proposals and introduced the paragraphs “discussion” and “future directions”.

Reviewer 2 Report

Comments and Suggestions for Authors

This is a concise article describing the key concepts and current data on biliary and gallbladder intervetion. The article is overall clear and well written. Images have an adequate quality. 

1-I suggest to provide mode evidence based on current studies, procedure success and complications. 

2-Images on complication after interventions can be provided if available. 

3-Povide the main indication when to perform the procedure. These can be also summarized in a dedicated table. 

Author Response

Dear Madam, dear Sir,

We appreciate your suggestions.  

1: We mentioned success rates of PGBD in the second sub-paragraph of “data background”. Complications of PGBD are low, drainage dislocation is the most common adverse event and is reported some paragraphs below.

2: We apologize, but we cannot provide images of complications after the procedure. An image of a spontaneous gallbladder perforation is part of the article.

3: We tried to outline the indications in the first paragraphs of PTBD and PGBD in the text. As there is a general indication for US in the primary puncture in PTBD and just one indication for PGBD (unfit for surgery and/or endoscopy), we did without a table.

Reviewer 3 Report

Comments and Suggestions for Authors

The manuscript discusses the role of ultrasonography in draining the biliary tract. It is written in good shape and illustrative. The ease of use and avoidance of radiologic exposure are of utmost importance. Also, the easier training and fewer complications are important privileges. This manuscript came late. Recently, EUS availability is a new imaging modality that competes with ultrasonography in intervention procedures, however, it is an invasive maneuver. Few comments:

I wonder if you can add arbitrary economic costs to these maneuvers and how they cost. This may help in deciding the choice of different procedures.

I have to say that some of these procedures are becoming historic such as cholecystostomy.

Adding a paragraph to compare with the newly advent procedure such as EUS concerning technique, complications, and cost will be of value.

Line 25-26: US is the only modality that allows continuous and real-time monitoring of the needle trajectory and the advancement of the needle tip through the penetrated tissue.------> This is not true!!! EUS can do the same, however, to be true, you can add without radiation exposure.

Comments on the Quality of English Language

Minor mistakes

Author Response

Dear Madam, dear Sir,

We appreciate your comments and suggestions.  

1: We can estimate the costs of the procedures in our country, but due to varying prices and reimbursement practises we are unable to give a statement that is reliable internationally. However, we introduced a paragraph on costs in the discussion.

2: Cholecystostomy might become historic in wealthy countries and prosperous health care systems. Today, it still is a frequently practised method around the world. Both authors perform EUS as well, so we agree that EUS will play a greater role in the future.

3: As you suggested, we added a paragraph on EUS-GBD addressing the suggested topics.

4: “US, whether applied percutaneously or endoscopically, is the only modality that allows continuous and real time monitoring of the needle trajectory and the advancement of the needle tip through the penetrated tissue.” This is what we wanted to say. We agree completely.

Round 2

Reviewer 1 Report

Comments and Suggestions for Authors

Dear authors

The requested corrections have been made, and now the paper is in a format according to the instructions for authors  and suitable for acceptance.

Congratulations.